

# *Lendatus*, a new genus of Xanthopygina (Coleoptera: Staphylinidae: Staphylininae) with description of three new species

Stylianos Chatzimanolis

Department of Biology, Geology and Environmental Science, University of Tennessee at Chattanooga, Chattanooga, TN, United States of America

## ABSTRACT

A new genus of Xanthopygina rove beetles is described here as *Lendatus* **gen. nov**. The new genus includes three new species: *L. bolivianus* **sp. nov.**, described from Bolivia, *L. philothalpiformis* **sp. nov.** described from Costa Rica and Panama, and *L. platys* **sp. nov.** described from Bolivia, Colombia, Ecuador and Peru. *Lendatus* belongs to the *Isanopus* group of genera of Xanthopygina and can distinguished from all the genera based on the unique punctation on the pronotum and the long apical setae of the paramere. A key to the three species of *Lendatus* along with photographs and illustrations is provided for the identification of species.

## INTRODUCTION

Xanthopygina is a diverse group of mostly neotropical rove beetles that includes (before the publication of this paper) 29 genera. In the latest phylogenetic analyses of the subtribe, *Chatzimanolis & Brunke (2019)* were able to examine all genera of Xanthopygina and identified the major lineages of the subtribe, One of them was the *Isanopus* group of genera, which included four genera: *Zackfalinus* Chatzimanolis (*Chatzimanolis, 2012*) as the sister group of *Peripus* Chatzimanolis & Hightower (*Chatzimanolis & Hightower, 2019*); identified in the phylogeny paper as genus 5), and *Isanopus* Sharp (*Chatzimanolis, 2008*) as the sister group of genus 2. That genus 2 is described in this paper as the new genus *Lendatus* Chatzimanolis and includes three new species.

The sister group relationship between *Isanopus* and *Lendatus* was first identified by *Chatzimanolis (2014)* in the first molecular phylogeny of the subtribe, where *Lendatus* was presented in that phylogeny as 'undescribed genus'. Delimiting new taxa, especially above the species level is not straightforward and ideally one should have multiple lines of evidence before proposing formal taxonomic names. While I had strong molecular evidence that *Lendatus* is indeed a new genus for quite some time, I did not feel comfortable describing *Lendatus* as new taxon until the completion of the morphological analysis of the subtribe that included all described genera and a number of undescribed ones.

Corresponding author
Stylianos Chatzimanolis,
stylianos-chatzimanolis@utc.edu

## MATERIALS & METHODS

Specimen preparation, study and photography followed other recently published papers on Xanthopygina (e.g., *Chatzimanolis & Hightower, 2019*). Dissected aedeagi were placed in small glass vials filled with glycerin and pinned underneath the specimen. I took the following measurements: HL: head length, at middle, from the anterior margin of frons to the nuchal ridge; HW: Head width, the greatest width, including the eyes; PL: pronotum length, at middle; PW: pronotum width, greatest width; EL: elytra length, measured in lateral view from the anterolateral angle of the elytra to the apex of the elytra; however, I used these measurements only proportionally (e.g., PW/PL). As a surrogate of total body length, I used forebody length (FL), measured by adding HL+PL+EL. I examined specimens using an Olympus ZX10 stereomicroscope and I took photographs using a Canon 40D camera equipped with a MP-E 65 mm macro lens on a Cognisys StackShot 3X macro rail and controller (https://www.cognisys-inc.com/products/stackshot/stackshot.php). I automontaged images using Helicon Focus Pro 6.7.1 (http://www.heliconsoft.com/heliconsoft-products/helicon-focus/). I removed the background of photographs using Fluid Mask 3 (https://www.vertustech.com). Type labels are separated by a slash '/'. Text within brackets [ ] is explanatory and was not included in the original label. Generic description was extracted from the matrix in *Chatzimanolis & Brunke (2019)* with addition of a few other characters. I produced maps using the online program SimpleMappr (*Shorthouse, 2010*). In this paper, I used the phylogenetic species concept of *Wheeler & Platnick (2000)* to delimit different species. Datasets for each species in DarwinCore format are available online at https://figshare.com/authors/Stylianos_Chatzimanolis/384794.

I examined specimens from the following institutions:

BMNH       The Natural History Museum, London, UK (M. Barclay).
CMNC       Canadian Museum of Nature, Ottawa, ON, Canada (R. Anderson).
CMNM       Carnegie Museum of Natural History, Pittsburgh, PA, USA (R. Davidson).
CNC        Canadian National Collection, Ottawa, ON, Canada (A. Brunke).
CRO        G. de Rougemont collection, Oxford, UK (G. de Rougemont).
DEBU       University of Guelph Insect Collection, Guelph, ON, Canada (S. Marshall).
FMNH       Field Museum of Natural History, Chicago, IL, USA. (C. Maier).
MNCR-A     National Museum of Costa Rica, San José, Costa Rica (A. Ruiz).
MUSM       Universidad Nacional Mayor de San Marcos, Museo de Historia Natural, Lima, Peru (D. Silva).
NHMD       Natural History Museum of Denmark, University of Copenhagen, Copenhagen, Denmark (A. Solodovnikov).
SEMC       Snow Entomological Collection, Biodiversity Institute, University of Kansas, Lawrence, KS, USA (Z. Falin).
UNSM       University of Nebraska State Museum, Lincoln, NE, USA (B. Ratcliffe).
UTCI       The University of Tennessee at Chattanooga, Chattanooga, TN, USA (S. Chatzimanolis).

Please note that several of the specimens currently deposited in SEMC will be transferred to MUSM per previous institutional/collecting agreements.

The electronic version of this article in Portable Document Format (PDF) will represent a published work according to the International Commission on Zoological Nomenclature (ICZN), and hence the new names contained in the electronic version are effectively published under that Code from the electronic edition alone. This published work and the nomenclatural acts it contains have been registered in ZooBank, the online registration system for the ICZN. The ZooBank LSIDs (Life Science Identifiers) can be resolved and the associated information viewed through any standard web browser by appending the LSID to the prefix http://zoobank.org/. The LSID for this publication is urn:lsid:zoobank.org:pub:0612FF19-38E8-4072-AF74-0EB16165841. The online version of this work is archived and available from the following digital repositories: PeerJ, PubMed Central and CLOCKSS.

# RESULTS

Taxonomy
*Lendatus* Chatzimanolis, new genus
(Figs. 1–9)
urn:lsid:zoobank.org:act:73EEC4F3-E35B-4E67-9FAA-5C8C14222ABB

**Type Species.** *Lendatus platys*, new species, here designated.
**Diagnosis**. *Lendatus* belongs to the *Isanopus* group of genera (see *Chatzimanolis & Brunke, 2019* for characters differentiating all genera in the *Isanopus* group) based on the following morphological characteristics: basal transverse carina on sternum 3 acutely pointed medially; lack of dense meshed microsculpture on sterna 5–7 (Fig. 4C); antennomeres 8–10 quadrate or elongate (Fig. 3E); and mesocoxae moderately to strongly separated (Fig. 4B). *Lendatus* was recovered as the sister group to *Isanopus* (*Chatzimanolis, 2014*; *Chatzimanolis & Brunke, 2019*) and the sister group relationship is supported by the following morphological characteristics (besides the molecular data supporting that relationship): coarse punctures impressed in flange at posterior angle of pronotum (Fig. 2); and lateral area of basal transverse carina on sternum 3 sinuate. Synapomorphies for *Lendatus* include: apical setae on paramere long, produced over the median lobe (Figs. 5–7), longer than any other Xanthopygina genus; and distribution of punctures on disc of pronotum split into anterior and posterior parts by diagonal longitudinal line, a unique character state in Xanthopygina. Additional characteristics that can distinguish *Lendatus* from *Isanopus* include: paramere not extremely reduced (as in *Isanopus*) and tarsomeres of middle and hind legs not enlarged and lobed (as in *Isanopus*).

Some species of *Oligotergus* Bierig may look superficially similar to *Lendatus*, but species in that genus typically lack the characteristics of the *Isanopus* group. Additionally, *L. philothalpiformis* has the same color pattern with some *Philothalpus* Kraatz species but *Philothalpus* can be easily distinguished by the presence of a pair of accessory ridges on the anterior basal transverse carina of tergum 3 (see *Chani-Posse et al., 2018*).

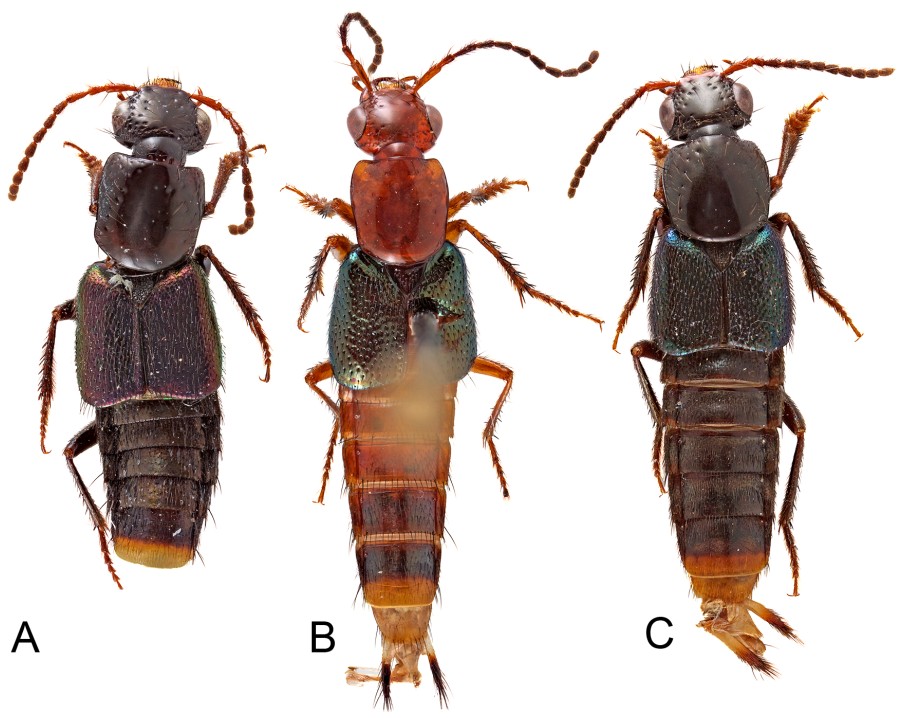

**Figure 1  Habitus photographs of species of *Lendatus* Chatzimanolis.**  (A) *Lendatus bolivianus* Chatzimanolis. (B) *Lendatus philothalpiformis* Chatzimanolis. (C) *Lendatus platys* Chatzimanolis. Not to scale.

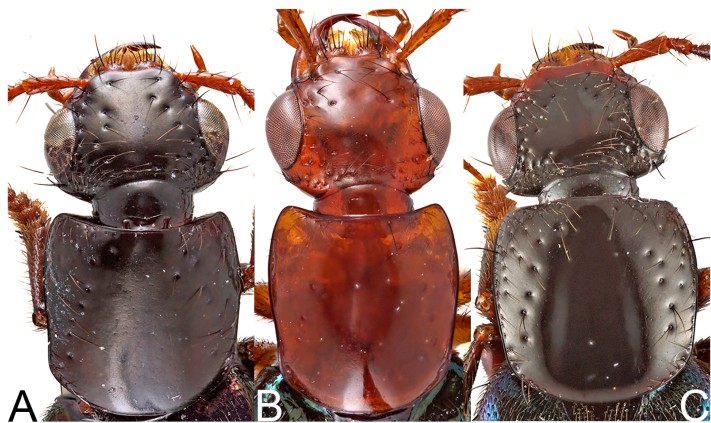

**Figure 2  Heads and pronota of species of *Lendatus* Chatzimanolis.**  (A) *Lendatus bolivianus* Chatzimanolis. (B) *Lendatus philothalpiformis* Chatzimanolis. (C) *Lendatus platys* Chatzimanolis. Not to scale.

**Description.** Habitus as in Fig. 1. Body medium-sized, forebody 4.6–5.8 mm long; without long bristle-like setae. Coloration of head and pronotum dark brown to black with metallic overtones or bright reddish-orange; elytra dark metallic green, blue or purple; abdomen dark brown or reddish brown to dark brown.

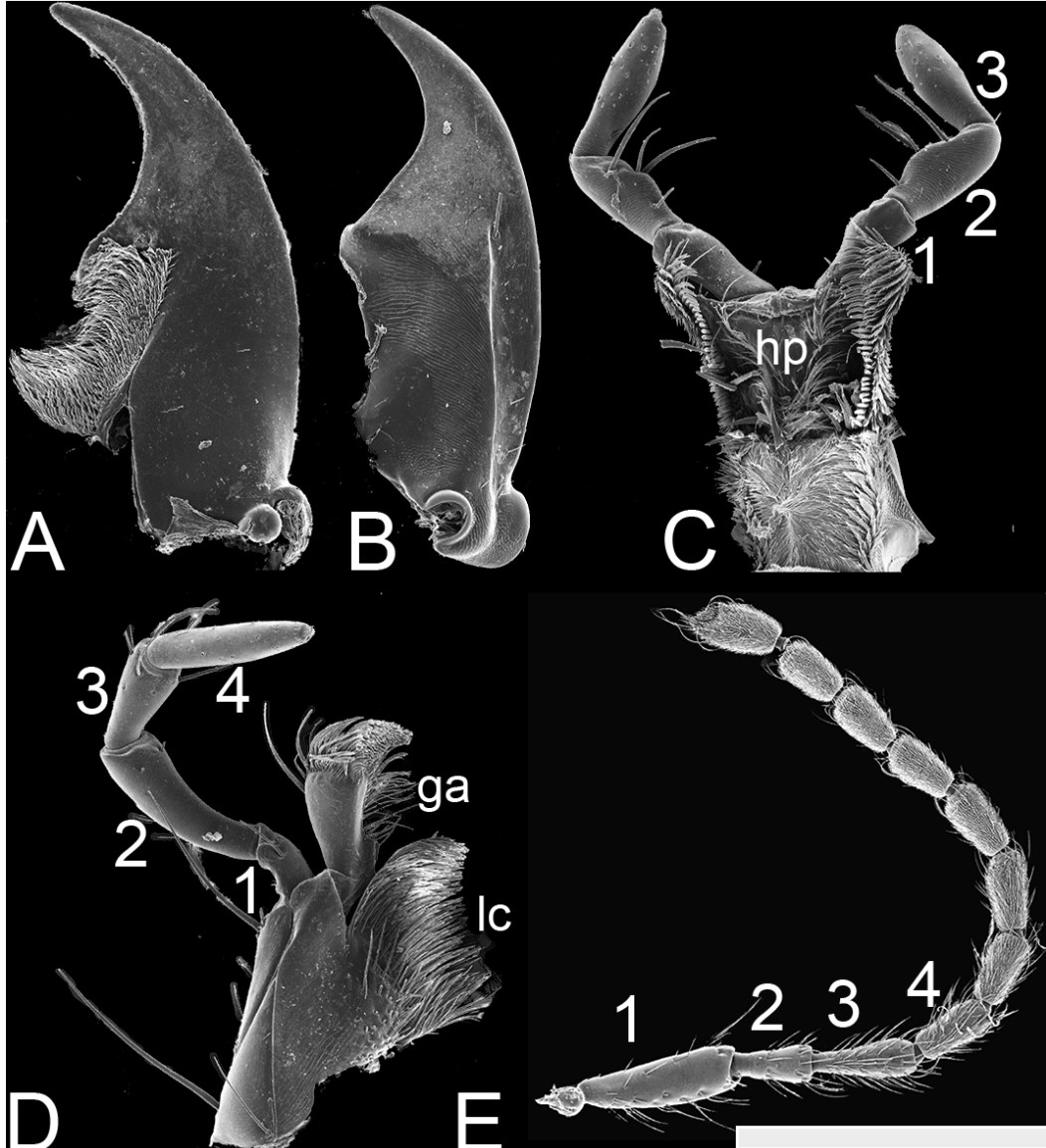

**Figure 3** **SEM photographs of *Lendatus philothalpiformis* Chatzimanolis.** (A) Ventral view of left mandible, scale bar = 0.56 mm. (B) Dorsal view of right mandible, scale bar = 0.56 mm. (C) Hypopharynx and labial palps, scale bar = 0.88 mm. (D) Maxilla, scale bar = 0.56 mm. (E) Antenna, scale bar = 1.09 mm. Numbers above the antenna, the maxillary palp and the labial palp correspond to the different segments; ga: galea; lc: lacinia; hp: hypopharynx.

Head (Fig. 2) shape rectangular; head length in comparison to pronotum shorter to subequal. Eye size relative to length of head large, more than 3/4 length of head. Postclypeus in comparison to frons not deflexed, anterior margin more or less straight. Middle of epicranium impunctate but with microsculpture. Postmandibular ridge laterally; with deep punctures demarcating raised postmandibular ridge dorsolaterally present. Gular

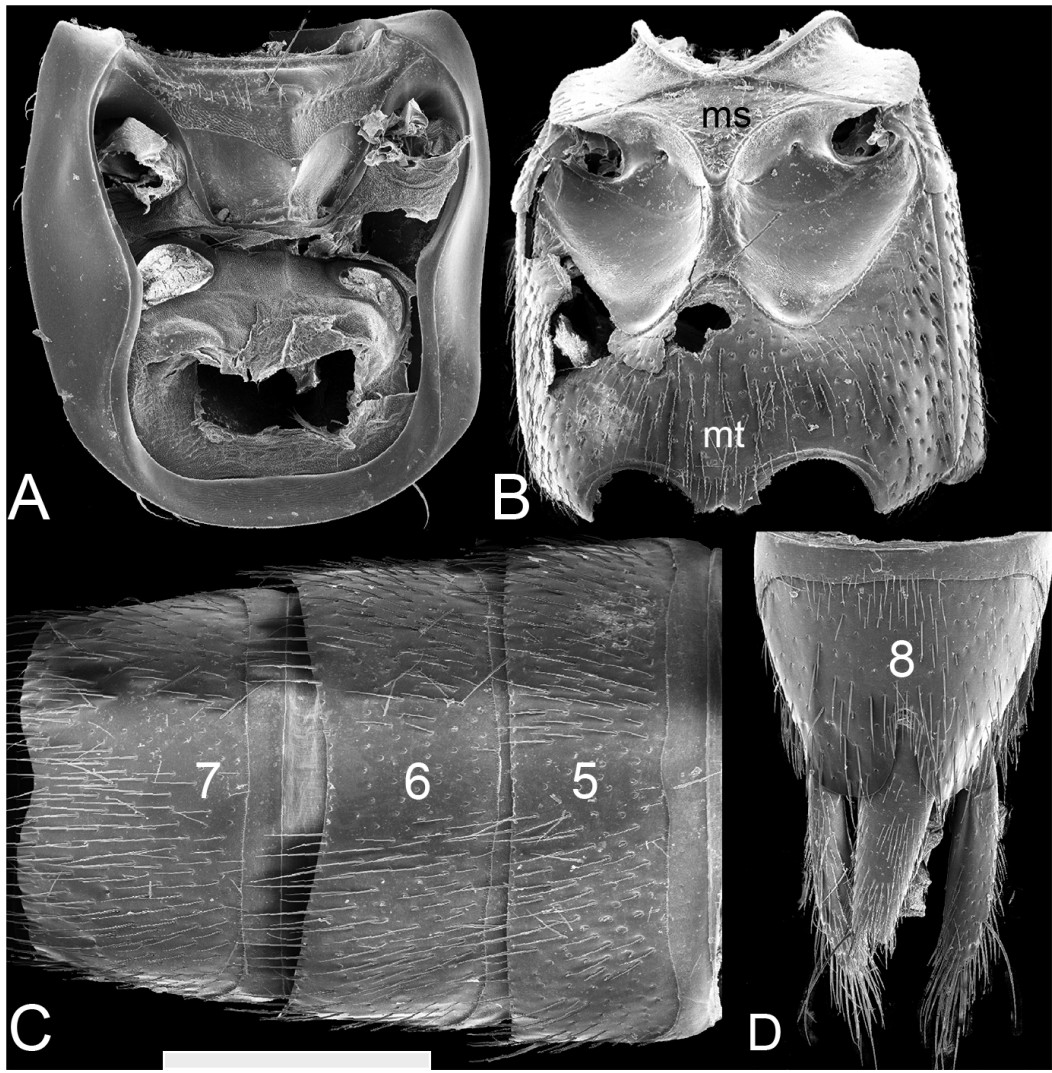

**Figure 4** **SEM photographs of *Lendatus philothalpiformis* Chatzimanolis.** (A) Prosternum and pronotal hypomeron, scale bar = 1.09 mm. (B) Meso- and metaventrite, scale bar = 1.44 mm. (C) Abdominal sterna 5–7, scale bar = 1.25 mm. (D) Abdominal sterna 8–9, scale bar = 1.27 mm. Numbers on the abdomen correspond to the number of segments; ms: mesoventrite; mt: metaventrite.

sutures not joined before neck extended close to each other at base of head capsule. Nuchal ridge present. Neck disc punctures sparse.

Antennae (Fig. 3E), antennomere 1 same width or slightly wider than 2. Antennomere 3 elongate, three times as long as wide; antennomere 4 with tomentose pubescence; antennomere 6 with curved, distinctly longer and thicker subapical setae than other macrosetae, forming circlet; antennomeres 1–11, cylindrical, longer than wide; antennomeres 8–10 symmetrical; antennomeres 5–10 without club; antennomere 11 in males subequal to 10.

Mouthparts with labrum having broad U-shaped emargination, lobes strongly separated. Mandibles (Figs. 3A–3B) relative length typical (i.e., closed mandible not extending beyond margin lateral margin of head); without asymmetrical torsion. Mandibles in dorsal view curved from apical half; in lateral view straight; left and right mandibles each with one tooth. Maxilla (Fig. 3D) with galea much shorter than palpus; maxillary palpus with $P_3$ distinctly shorter than $P_2$; $P_4$ distinctly longer than $P_3$; $P_4$ not dilated. Hypopharynx and labial palpi as in Fig. 3C; labial palpus $P_3$ widest before apex, without long dense setae on entire lateral sides. Ligula small, entire. Mentum with alpha setae present; hypostomal cavity present, moderately delimited.

Pronotum (Fig. 2) shape of lateral margins in dorsal view posteriad of midpoint straight to sinuate (except *L. platys* convex); anterior angles in dorsal view not strongly acuminate and produced laterad. Pronotum near anterolateral angles without raised impunctate spots; anterolateral corners with punctation; disc of pronotum with punctation split into anterior and posterior parts by diagonal longitudinal line of punctures; with coarse punctures impressed in flange at posterior angle of pronotum; with microsculpture. Pronotum subquadrate; narrower than head at widest points. Hypomeron (Fig. 4A) with superior marginal line continuous to anterior margin; superior marginal line without deflection under anterior angles in ventral view; inferior marginal line continued as a separate entity beyond anterior pronotal angles and curving around them. Postcoxal process absent. Basisternum slightly longer than furcasternum; basisternum with pair of macrosetae situated far from anterior margin of prosternum.

Elytra without contiguous polygon-shaped meshed microsculpture or patches of white setae. Elytral setae not reduced, easily seen at low magnification (e.g., 40x). Mesoventrite (Fig. 4B) with anterior margin forming "lip"; without median carina; mesoventral process triangular; process extended distally to distance about 2/5 between mesocoxae. Metaventrite (Fig. 4B) with large punctures; metaventral processes, small, rounded, triangular, extended to beginning of metacoxae.

Legs with tarsal segmentation 5-5-5; prefemora without lateroventral apical spines; protarsi with modified pale (adhesive) setae ventrally; tarsomeres 1–4 of protarsi dorsoventrally flattened. Mesocoxae (Fig. 4B) moderately separated; intercoxal area distinctly recessed compared to mesoventrital process. Metacoxae without coxal shield; metatibia without thick and long apical spurs but with smaller spurs and spines. Meso/metatarsi without asymmetrically lobed tarsomeres 1–4; tarsomeres 3–5 of metatarsi with chaetotaxy developed only at margins of dorsal surface, dorsal surface of tarsomeres glabrous along midline. Pretarsal claws with empodial setae.

Abdomen (Figs. 4C–4D) with protergal glands having well-developed acetabula. Anterior basal transverse carina on terga 3–5 without pair of accessory ridges; tergum 3 without posterior basal transverse carina and without curved carina (arch-like) on disc; center of tergum 5 with punctation; posterior half of tergum 5 in lateral view not appearing bulging. Sternum 3 with acutely pointed basal transverse carina medially; laterally basal transverse carina sinuate; basal transverse carina absent on sternum 4; sternum 5 without dense, meshed microsculpture anterolaterally; sternum 7 with sparse punctation laterally. Males with secondary sexual structures (emargination medially on sterna 7 and 8); without porose structure. Females without obvious secondary sexual structures.

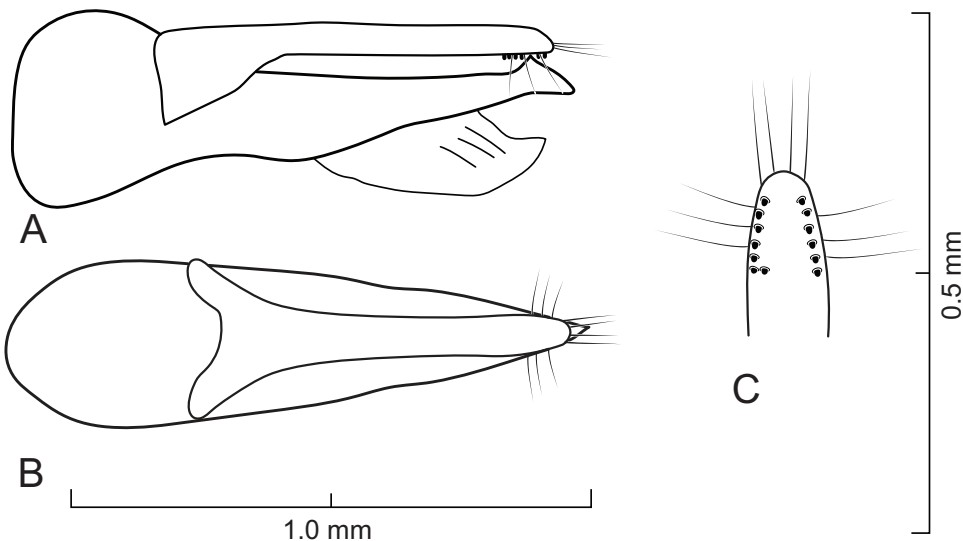

**Figure 5** **Aedeagus of *Lendatus bolivianus* Chatzimanolis.** (A) Lateral view. (B) Dorsal view. (C) Detail of paramere, ventral view.

Aedeagus as in Figs. 5–7; with long median lobe and single paramere; paramere with sensory peg setae and long apical setae; median lobe with single subapical tooth; median lobe without apical tooth, carina or paired apex. Spermatheca not sclerotized.

**Etymology.** The name is in honor of my dear friends Dr. Ntina Karametsi, Dr. Lia Koutelou, Mr. Dimitris Kotsis, Dr. Tania Patsialou and Dr. Eleni Zika. The name is made up from a combination of letters from the first names. The name is masculine.

**Habitat.** Collected in lowland tropical rainforests and mid-elevation cloud forests using a variety of trapping techniques and by shifting leaf litter. The genus most likely inhabits the leaf litter.

### *Lendatus bolivianus* Chatzimanolis, new species
(Figs. 1A, 2A, 5, 8)
urn:lsid:zoobank.org:act:14E6C64D-E882-41D3-85DA-75930F62DCF1

**Type material.** **Holotype**, here designated, male, "Bolivia: La Paz, 9.4 km E. Chulumani, Apa-Apa, 2,400 m, 16°20.99S 67°30.30W [−16.349833, −067.505], 17.i.2001, R. Anderson, upper yungas litter, BOLA01-002" / "SM0459200 [barcode label]" / "HOLOTYPE *Lendatus bolivianus* Chatzimanolis, des. Chatzimanolis 2019". In the collection of SEMC.

**Paratypes.** Six; one with same locality label as holotype and barcode label SM0459190 (1 ♀ SEMC); "Bolivia: La Paz Prov. Chulumani, 9.2 km E of, 2300 m, 16°20.59S 67°30.18W [−16.3431667, −67.503], 19–21 Jan[nuary] 2001, J. S. Ashe, R. S. Hanley, BOL1AH01 039 ex: flight intercept trap" / "SM0236239" (1 ♂ SEMC); "Bolivia: La Paz 9.4 km E Chulumani, 2,200 m, 16°20.99S 67°30.30W [−16.349833, −067.505], 19–21.i.2001, J. S. Ashe, R. S. Hanley, BOL1AH01 038 ex: flight intercept trap" / SM0574084, SM236231 (1 ♀ SEMC; 1 ♀ UTCI ); "Bolivia: Chulumani, Apa-Apa forest, 16°21′S, 67°30′W [−16.35,

−67.5], 12–14.xi.2007, 2,000 m, shifting forest litter, V. Grebennikov leg." (1 ♀, 1 ♂ NHMD). All paratypes with label "PARATYPE *Lendatus bolivianus* Chatzimanolis, des. Chatzimanolis 2019".

**Diagnosis.** *Lendatus bolivianus* and *L. platys* can be distinguished from *L. philothalpiformis* by the coloration of head and pronotum (dark brown to black in *L. bolivianus* and *L. platys*; bright reddish-orange in *L. philothalpiformis*). *Lendatus bolivianus* can be distinguished from *L. platys* by the shape of the pronotum (becoming narrower (concave) posteriorly (Fig. 2A) in *L. bolivianus*; becoming wide (convex) posteriorly (Fig. 2C) in *L. platys*); the shape of the paramere (paramere wider, converging to apex in dorsal view (Fig. 5B) in *L. bolivianus*; paramere narrower, parallel-sided from base to apex in dorsal view (Fig. 7B) in *L. platys*); and the length comparison between the anterior portion of the paramere and median lobe (median lobe slightly longer than paramere (Figs. 5A–5B) in *L. bolivianus*; median lobe much longer than paramere (Figs. 7A–7B) in *L. platys*).

**Description.** Forebody length 4.9–5.5 mm. Coloration of head, pronotum and ventral side of body dark brown to black; mouthparts and antennae dark orange; elytra metallic purple with green overtones; legs dark brown except tarsi dark orange; abdomen dark brown to black except segment 7 (posterior 1/4 orange) and segment 8 (orange).

Head with 1–2 irregular rows of medium-sized punctures on each side of central impunctate area (except anteriorly); with additional 3–4 large punctures on epicranium; with microsculpture and micropunctures. Head width/length ratio = 1.61. Pronotum width/length ratio = 0.95; pronotum widest anteriorly, becoming gradually narrower posteriad; diagonal longitudinal line of punctures on disc of pronotum with 3–4 large punctures; anterolateral to that line pronotum with 5–6 medium-sized punctures; posterolateral to that line pronotum impunctate; pronotum with microsculpture and sparse micropunctures; pronotum/elytra length ration = 0.82. Males with narrow, deep emargination on sternum 7; sternum 8 with deep U-shaped emargination.

Aedeagus as in Fig. 5; paramere in dorsal view gradually converging to rounded apex; in lateral view paramere slightly convex, converging to broadly rounded apex; paramere with peg setae as in Fig. 5C; paramere narrower but slightly longer than median lobe; median lobe in dorsal view converging to apex; in lateral view median lobe becoming narrower from middle to apex; with small dorsal subapical tooth.

**Distribution.** Known from the province of La Paz in Bolivia.

**Habitat.** All specimens were collected in the Yungas forest along eastern slope of the Andes Mountains in Bolivia (at elevations of 2,000 m or above) by shifting litter or flight intercept traps.

**Etymology.** The specific epithet refers to the country of Bolivia.

*Lendatus philothalpiformis* **Chatzimanolis, new species**
(Figs. 1B, 2B, 3, 4, 6, 9)
urn:lsid:zoobank.org:act:7AFD3EE5-49B1-495D-A289-2C390B06BF61

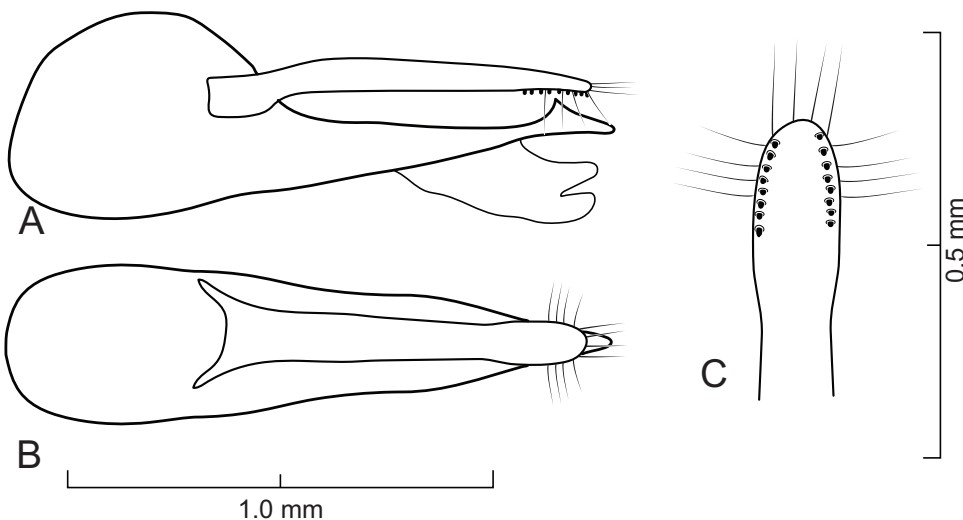

**Figure 6** **Aedeagus of *Lendatus philothalpiformis* Chatzimanolis.** (A) Lateral view. (B) Dorsal view. (C) Detail of paramere, ventral view.

**Type material**. **Holotype**, male, here designated, "Costa Rica: Puntarenas, Corcovado National Park, Sirena Station, upper Rio Claro trail, 100 m, 8°28′29″N 83°35′8″W [8.474722, −83.58555], 28.Jun[e]–1.Jul[y].2000, Z.H. Falin, CR1ABF00 061, ex: flight intercept trap" / "SM0203906 [barcode label]" / "HOLOTYPE *Lendatus philothalpiformis* Chatzimanolis, des. Chatzimanolis 2019". In the collection of SEMC.

**Paratypes.** 121: "Costa Rica: Alajuela, Estac. Biol. San Ramón, 900 m, 1.vii.–31.viii.1995, P. Hanson, CR1H93-95 5, ex: malaise trap" / "SM0075968" (1 ♂ SEMC); same locality except 1.viii–30.ix.1995, CR1H93-95 6, SM0075818 (1 ♀ SEMC); same locality except 10°13′4″N 84°35′46″W [10.21777, −84.596111], xi.–xii.1999, SM0457580 (1 ♂ SEMC); same locality except 10°13′4″N 84°35′46″W [10.21777, −84.596111], ii.–iii.2000, SM0457607 (1 ♀ SEMC); "Costa Rica: Alajuela, E.B. San Ramón, R.B. San Ramón, 27 km N & 8 km W San Ramón, 10°13′30″N 84°35′30″W [10.225, −84.591666], 850–950 m, 29.vi.–6.vii.1999, R. Anderson, wet premontane forest CR1A99-108A" / "SM0188194" (1 ♂ SEMC); same locality except 900 m, CR1A99-113B, SM0186510 (1 ♀ SEMC); same locality except 810 m, 10°13′4″N 84°35′46″W [10.21777, −84.596111], 8.vii.2000, J.S. Ashe, R. Brooks, Z.H. Falin, CR1ABF00 084, ex: flight intercept trap, SM0203647, SM0203665 (1 ♂, 1 ♀ SEMC); same locality except 900 m, 10°13′4″N 84°35′46″W [10.21777, −84.596111], 8.vii.2000, P. Hanson, CR1EH99 01, SM0235433 (1 ♂ SEMC); "Costa Rica: Prov. Alajuela, A.C.A. San Ramón, Reserva Biol Alberto Brenes, Rio San Lorencito, 850 m, 24.iii.1999, C. Moraga, Sombrereta, L_N_245500_470800 #52477" / INB0003030776, INB0003030777, INB0003030779 (2 ♂, 1 ♀ NHMD); "Costa Rica: Prov. Alajuela, San Ramón, Est. Biol. Villa Blanca, Send. La Capilla, 1,115 m, 16.iii.–9.iv.2010, B. Hernández, Tp. Malaise, L_N_242482_483371 #99630" / "INB0004248707" (1 ♀ NHMD); "Costa Rica: Prov. Alajuela, Upala, P.N. Volcán Tenorio, Cerro La Carmela, 1,026 m, 17.ii.–18.iv.2010, J.A. Azofeifa, Tp. Malaise, L_N_298828_427338 #99732" "INB0004256029" (1 ♀ NHMD);

"Costa Rica: Alajuela, Peñas Blancas, 800 m, 19.v.1999, J.S. Ashe, R. Leschen, R. Brooks, ex: flight intercept trap" / "SM0046201" (1 ♀ SEMC); "Costa Rica: Prov. Alajuela, La Fortuna, Sector Catarata, 500 m, 3.xi.1997–6.i.1998, G. Garballo, Malaise, L_N_268500_462500 # 48837" / "INBIOCR002595077" (1 ♀ MNCR-A); "Costa Rica: Cartago Prov., Refugio Nac. de Fauna Silvestre Tapanti, 2 km E Station, 1,320 m, 9°44.287′N83°46.875′W [9.738116, -83.78125], 30.x.–1.xi.2001, R. Brooks, ex: flight intercept trap, CR1B01 15" / SM0474732, SM0474730, SM0474731, SM047429 (2 ♂, 1 ♀ SEMC; 1 ♂ UTCI); same locality except 1 km E Station, 1,410 m, 9°45.129′N 83°46.936′W [9.75215, −83.782266], CR1B01 13, SM0474724 (1 ♀ SEMC); "Costa Rica: Prov. Cartago, La Represa. Tapanti, 1800 m, vii.1995, R. Delgado, interseccion LN 185900 563300 #5342" / "INBIOCR002209951" (1 ♂ MNCR-A); "Costa Rica: Prov. Cartago, Pejibaye, Estación Biológica Copal, Sendero Tigra, 1,090 m, 3–14.iv.2005, J. Azofeira Z., Tp. Malaise, L_N_196286_563684 #80039" / "INB0003938486" (1 ♂ NHMD); "Costa Rica: Guanacaste, Guanacaste Conservacion Area, Maritza Field Station, 950 m, 13.ii.1996, R. Anderson, CR1A96 010C, ex: dry-tropical wet forest trans. litter" "SM0083887" (1 ♂ SEMC); "Costa Rica: Guanacaste, Estac. Cacao, 1000–1400 m, SW side Volcan Cacao, vii.1989–iii.1990, Malaise, TP.-GNP Biod. Survey" / INBIOCR000203134, INBIOCR000248458, INBIOCR000258332, INBIOCR000203124, INBIOCR000203105, INBIOCR000168862 (2 ♂, 4 ♀ MNCR-A); same locality label except II curso Parataxon., vi.1990, INBIOCR000250397 (1 ♂ MNCR-A); same locality label except iii–viii.1990, INBIOCR000231448 (1 ♀ MNCR-A); same locality label except 21–29.v.1992, INBIOCR000374813 (1 ♀ MNCR-A); same locality label except 1988–1989, INBIOCR000101546, INBIOCR000042128 (2 ♀ MNCR-A); "Costa Rica, Guanacaste, Estac. Pitilla, 9 km S Santa Cecilia, 700 m, xi.1989, C. Moraga & P. Rios, 330200, 380200" / "INBIOCR000111406" (1 ♂ MNCR-A); "Costa Rica, Guanacaste, Tierras Morenas, 685 m, xi.1993, G. Rodriguez, L N 287800_427600 #2476" / "INBIOCR001947013" (1 ♀ MNCR-A); "Costa Rica, Prov. Guanacaste, Macizo Miravalles, Estac. Cabro Muco. Sitio Azufral, 1,100 m, 22.ix.–5.x.2003, J. Azofeifa, Intersección L_N_299769_411243 #75479" / "INB0003771446" (1 ♀ NHMD); "Costa Rica: Heredia Prov., 6 km ENE Vara Blanca, 10°11′N 84°07′W [10.18333, −84.11666], 1,950 m, 15–22.iv.2002, montane forest leaf litter, R. Anderson, CR2A02 03" / SM0527314, SM0527301 (2 ♂ SEMC); "Costa Rica, Heredia, Finca Murillo, 9 km NE Vara Blanca, 1,450–1,550 m, 10°14′17″N 84° 06′06″W [10.238055, −84.101666], R. Anderson, 14–20.ii.2005, INbio-CET-ALAS transect, CRA105 007" / "SM0693946" (1 ♀ SEMC); "Costa Rica: [Heredia] Vara Blanca, viii.[19]38" / "Field Mus. Nat. Hist.1966, A. Bierig Colln., Acc. Z-13812" (1 ♂ FMNH); "Costa Rica: Prov. Limón, P.N. La Amistad. Punto., 1,300–1,400 m, 25.x.–2.xi.2007, M. Moraga, B. Gamboa, Tp. Malaise, L_N_198990_627455 #92615" / "INB0004126042" (1 ♂ NHMD); "Costa Rica: Prov. Limón, Manzanillo, RNFS Gandoca y Manzanillo, 0–100 m, 9.xi.–13.x.1992, K. Taylor, L-S 398100, 610600" / "INB000937676" (1 ♀ MNCR-A); "Costa Rica: Puntarenas, Corcovado National Park, Sirena Station, Corcovado trail, 150 m, 8°29′7″N 83°34′39″W [8.485277, −83.57750], 28.Jun[e]–1.Jul[y].2000, Z.H. Falin, CR1ABF00 059, ex: flight intercept trap" / "SM0203552" (1 ♀ SEMC); "Costa Rica: Puntarenas, Corcovado National Park, Sirena Station, Rio Pavo trail, 5 m, 8° 29′5″N 83°35′33″W [8.484722, −83.5925],

25–28 Jun[e].2000, Z.H. Falin, CR1ABF00 037, ex: flight intercept trap" / "SM0203763" (1 ♂ SEMC); "Costa Rica: Puntarenas, Monteverde,, 24.v.1989, 1,400 m, J.S. Ashe, R. Leschen, R. Brooks, #419, ex: pitfall trap" / "SM0046200" (1 ♂ SEMC); same locality label except Boehme house, #437, SM0046209 (1 ♂ SEMC); same locality label except Cerro Chomogo, 1,550 m, flight intercept trap, SM0046211 (1 ♂ SEMC); same locality label except 1,520 m, flight intercept trap, SM0046199 (1 ♂ SEMC); same locality label except 1570 m, 9.v.1989, flight intercept trap, SM0046198 (1 ♀ SEMC); same locality label except 1630 m, 7.vii.1990, S.E. Roberts, flight intercept trap, SM0046193 (1 ♂ SEMC); same locality label except 1610 m, 7.vii.1990, S.E. Roberts, flight intercept trap, SM0046210, SM0046208 (2 ♂ SEMC); same locality label except 21.v.1989, flight intercept trap, SM0046195, SM0046204, SM0046202, SM0046197, SM0046194, SM0046205, SM0046196 (3 ♂, 2 ♀ SEMC; 1 ♂1♀ UTCI); same locality label except 1550 m, flight intercept trap, SM0046203 (1 ♂ SEMC); same locality label except 28–31.v.1992, M.L. Jameson, flight intercept trap, SM0045890 (1 ♂ SEMC); "Costa Rica, Puntarenas, San Luis-Monteverde, LN250-850-449-250, 17–31.xii.1993, Z. Fuentes, 1040 m, ex: malaise trap, #2583" / "SM0068168" (1 ♂ SEMC); same locality label except ii.1993, #1897, SM0068198, INBIOCR002522864, INBIOCR002522865, INBIOCR001166927 (1 ♀ SEMC; 3 ♀ MNCR-A ); same locality label except ii.1992, INBIOCR000842619 (1 ♀ MNCR-A); same locality label except 1,000–1,350 m, 17–31.xii.1992, #2583, INBIOCR002523162 (1 ♂ MNCR-A); same locality label except 1,000–1,350 m, xii.1993, #2493, INBIOCR001714070 (1 ♂ MNCR-A); same locality label except vii.1993, #2424, INBIOCR002523005 (1 ♂ MNCR-A); same locality label except 1–31.x.1993, #2425, INBIOCR001957088 (1 ♂ MNCR-A); same locality label except vii.1992, INBIOCR000722993 (1 ♂ MNCR-A); same locality label except x.1993, #2428, INBIOCR002523051 (1 ♂ MNCR-A); same locality label except xi.1993, #2443, INBIOCR001938006, INBIOCR001938032, INBIOCR001938005, INBIOCR001938033 (2 ♂, 2 ♀ MNCR-A); same locality label except ix.1993, #2429, INBIOCR002523059 (1 ♂ MNCR-A); same locality label, A. C. Arenal, xi.1993, #2427, Z. Fuentes, Amarilla, SM0068204, SM0068201, INBIOCR002523429, INBIOCR002523428, INBIOCR002523427 (2 ♂ SEMC; 1 ♂, 2 ♀ MNCR-A); same locality label, A. C. Arenal, i.1993, Z. Fuentes, LN 449250_250850 #2584, SM0068203, INBIOCR002523178, INBIOCR002523177, INBIOCR002523179 (1 ♂ SEMC; 3 ♂ MNCR-A); same locality label, A. C. Arenal, i.1993, Z. Fuentes, LN 449250_250850 #2585, SM0068196, SM0068200 (2 ♀ SEMC); same locality label except 20–27.vi.1994, #3029, INBIOCR001922841, INBIOCR001922842 (1 ♂, 1 ♀ MNCR-A); "Costa Rica, Puntarenas, Res. Biol. Monteverde, Est. La Casona, 1520 m, K. Flores, iv.1992, L-N 253250 449700" / INBIOCR000990559, INBIOCR000793519 (1 ♂, 1 ♀ MNCR-A); same locality label except ix.1991, INBIOCR000510117 (1 ♂ MNCR-A); "Costa Rica: Puntarenas Prov., Hacienda La Amistad, 8°58.102′N 82°46.883′W [8.968366, −82.781383], 1,900 m, premont.-lower mont. moist forest, sifting leaf litter, 12.vi.2012, Solodovnikov, Brunke, Puliafico, Selvantharan" / "Chatzimanolis DNA Voucher, Extraction SC-405, Extraction date: 27.iii.2015" (1 ♂ NHMD); "Costa Rica, Puntarenas, R.F. Golfo Dulce, 3 km SW Rincon, 10 m, v–vi.1992, P. Hanson, ex: malaise" / "SM0069525" (1 ♂ SEMC); "Costa Rica, Puntarenas, Altamira Biol. Sta. 1,510–1,600

m, 9°1.76′N 83°0.49′W [9.029333, −83.008166], 4–7.vi.2004, J. S. Ashe, Z.H. Falin, I. Hinojosa, ex: flight intercept trap, CR1AFH04 144” / “SM0606679” (1 ♂ SEMC); “Costa Rica, Puntarenas, Las Alturas Biol. Sta. 1,660 m, 8°56.17′N 82°50.01′W [8.936166, −82.8335], 31.v.–3.vi.2004, J. S. Ashe, Z.H. Falin, I. Hinojosa, ex: flight intercept trap, CR1AFH04 092” / “SM0606867” (1 ♀ SEMC); “Costa Rica: Puntarenas, Fca. Cafrosa, Est. Las Mellizas, P.N. Amistad., 1,300 m, R. Delgado, 19.vi.–26.vii.1990, L-S-316100, 596100” / “INBIOCR000667816” (1 ♂ MNCR-A); “Costa Rica, San Jose, Zurqui de Moravia, 1,600 m, iv.1994, P. Hanson, ex: malaise” / “SM0069535” (1 ♂ SEMC); same locality except iii.1994, SM0069520 (1 ♂ SEMC); same locality except 1–30.viii.1995, CR1H93-95 14, SM0077306 (1 ♂ SEMC); same locality except 10°3′0”N 84°1′0″W [10.05, −84.01666], 1–30.ix.1995, CR1H95-96 07, SM0134461 (1 ♂ SEMC); “Costa Rica: San Vito de C. B., Las Cruces, 1200 m, 9.vii.–7.viii.1982, malaise tr, B. Gill” (1 ♀ CNC); “Panama: Bocas del Toro, 4 km N. Boquete, La Culebra trail, 1,500 m, 17.vii.1995, A. Gillogly” / SM0004684, SM0004685 (1 ♂, 1 ♀ SEMC); “Panama: Bocas del Toro, 8°34N 81°50W [8.56666, −81.83333], 1,500 m, 25 km NNE San Felix, leg. J. Wagner, 6.vi.1980” / “FM(HD)#80-5, Berlese floor litter & root mat, nr. ridge top, Qda. Alicia cloud forest” (1 ♀ FMNH); same locality except 10–12.vi.1980, Camino I7, malaise trap (1 ♀ FMNH); “Panama: Chiriqui Prov., La Fortuna Cont. Divide Trail, 8° 46′N 82°12′W [8.766666, −82.2], 1,150 m, 23.v.–9.vi.1995, J. Ashe, R. Brooks, #155, ex: flight intercept trap” / SM0046213, SM0007044, SM0003729, SM0046214 (3 ♂, 1 ♀ SEMC); same locality except 1,100 m, #157, SM0003687 (1 ♀ SEMC); same locality except, 9–12.vi.1995, #185, SM0003498 (1 ♀ SEMC); same locality except 1,200 m, 9.vii.1995, R. Anderson, PAN2A95 10C, ex: berlese forest litter, SM0037220 (1 ♀ SEMC); same locality except Hydrolog. Trail, 8°42′N 82° 14′W, 1,050 m, 9–12.vi.1995, #188, SM0005052 (1 ♂ SEMC); same locality Hydrolog. Trail, 8°42′N 82°14′W [8.7, −82.233], 1,150 m, 23.v.–9.vi.1995, #156, SM0003747 (1 ♀ SEMC); “Panama: Chiriqui, 4 km N Sta. Clara Hartmann’s Finca, 27.vi.–3.vii.1981, B. Gill, 1,500 m” (1 ♀ CMNC); “Panama: Chiriqui, La Fortuna Dam, 1,200 m, 14.vi.–15.viii.1982, wet forest FIT, B. Gill” (1 ♀ CNC); All paratypes with label “PARATYPE *Lendatus philothalpiformis* Chatzimanolis, des. Chatzimanolis 2019”.

**Diagnosis.** *Lendatus philothalpiformis* can be easily recognized among the existing species in the genus due to the bright reddish-orange coloration of the head and pronotum. Additionally, it is the only species known with a Central American distribution.

**Description.** Forebody length 4.6–5.8 mm. Coloration of head, pronotum and prosternum bright reddish-orange (in few specimens brown); mouthparts, antennae and legs reddish-orange to brown; elytra metallic green or blue; meso- and metaventrite brown; abdomen reddish-orange to brown (frequently with segment 6 dark brown) except segment 7 dark brown with posterior 1/3 orange and segment 8 orange.

Head with 1–2 irregular rows of large punctures on each side of central impunctate area (except anteriorly); with additional 3–4 large punctures on epicranium; with microsculpture and micropunctures. Head width/length ratio = 1.5. Pronotum width/length ratio = 0.92; pronotum widest anteriorly, becoming strongly narrower (concave) posteriad; diagonal longitudinal line of punctures on disc of pronotum with 3–4 large punctures; anterolateral

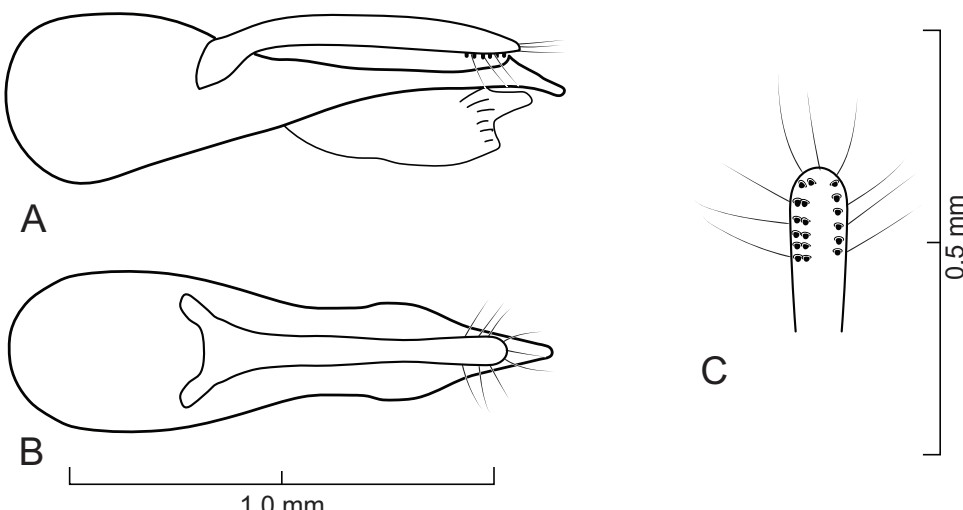

**Figure 7** **Aedeagus of *Lendatus platys* Chatzimanolis.** (A) Lateral view. (B) Dorsal view. (C) Detail of paramere, ventral view.

to that line pronotum with less than 5 large punctures; posterolateral to that line pronotum impunctate; pronotum with microsculpture and micropunctures; pronotum/elytra length ration = 0.92. Males with broad, shallow margination on sternum 7 (Fig. 4C); sternum 8 with shallow V-shaped emargination.

Aedeagus as in Fig. 6; paramere in dorsal view almost parallel-sided but apex wider; in lateral view paramere slightly convex, converging to narrow rounded apex; paramere with peg setae as in Fig. 6C; paramere narrower than median lobe except just before apex; paramere longer than median lobe; median lobe in dorsal view converging to apex; in lateral view median lobe becoming narrower from middle to narrowly elongate apex; with large dorsal subapical tooth.

**Distribution.** Known from many provinces in Costa Rica and the provinces of Bocas del Toro and Chiriqui in Panama.

**Habitat.** Specimens were collected with malaise, pitfall and flight intercept traps and by shifting leaf litter in wet tropical lowland forests or tropical cloud forests at elevations of 5–1,950 m).

**Etymology.** The specific epithet is derived from the words *Philothalpus* and *formis* and refers to the superficial resemblance of this species to species in the genus *Philothalpus*.

*Lendatus platys* **Chatzimanolis, new species**
(Figs. 1C, 2C, 7, 8)
urn:lsid:zoobank.org:act:920D79A5-D1D4-4B73-8263-30D4908E3823

**Type material**. **Holotype**, here designated, male, "Ecuador, Sucumbios, Sacha Lodge, 0.5°S 76.5°W [−0.5, −76.5], 270 m, 13–23.vi.1994, Hibbs, ex: malaise" / "SM0022600 [barcode

label]" / "HOLOTYPE *Lendatus platys* Chatzimanolis, des. Chatzimanolis 2019". In the collection of SEMC.

**Paratypes.** 48: same locality label as holotype, SM0022371 (1 ♂ SEMC); same locality label as holotype except 14–24.v.1994, SM0023298 (1 ♀ SEMC); same locality label as holotype except 3–16.viii.1994, SM0020931 (1 ♀ SEMC); "Bolivia: Santa Cruz, Amboro National Park, Los Volcanes, c.1,000 m, 18°06′S 63°36′W [−18.1, −63.6], 20.xi.–12.xii.2004" / "flight interception trap, H. Mendel & M.V.L. Barclay, BMNH(E) 2004-280" (5 ♂, 2 ♀ BMNH); "Ecuador: Morona-Santiago, Macas, 1,300 m, 20.ix.1989, M. Cooper" / "M. Cooper BMNH(E) 2004-275" (1 ♀ BMNH); "Ecuador: Napo, Yuturi Lodge, Rio Napo, 270 m, 0° 32′54″S 76°2′18″W [−0.548333, −76.03833], 20–21.iii.1999, R. Brooks, D. Brzoska, ECU1B99 010, ex: flight intercept trap" / SM0153450, SM0153439, SM0153432, SM0153459 (3 ♂ SEMC; 1 ♂ UTCI); "Ecuador: Napo, Tena-Baeza Rd. km 24, N. Cotundo, 36–4,000′, 3.v.1982, H. Frania, leaf litter, ridge" (1 ♂ FMNH); "Ecuador, Napo Prov. Yasuni N.P., Yasuni Research Sta., 0°38′S 76°36′W [−0.6333, -76.6], 215 m, 27.vii.–1.viii.1998, lowland rainforest, Ratcliffe, Jameson, Smith, Villatoro (1 ♂ UNSM); "Ecuador: Napo, Yasuni Nat. Park Biol. Res. Station, 220 m, 0.67°S 76.39°W [−0.67, −76.39], 18–26.v.1996, P. Hibbs, MT, primary forest (1 ♀ CNC); "Ecuador: Prov. Orellano, Yasuni Natl. Park, Yasuni Research Stn., 0°40′50″S 76°24′2″W [−0.680555, −76.400555], 250 m, 28.iv.–8.v.2009, on lead, M. Cannon (1 ♂ DEBU); "Ecuador: Sucumbios, Sacha Lodge, 270 m, 0°28′14″S 76°27′35″W [−0.470555, −76.45972], 21–24.iii.1999, R. Brooks, ECU1B99 047, ex: flight intercept trap" / SM0153262, SM0153263, SM0153255 (3 ♂ SEMC); "Ecuador: [Sucumbios], Napo R. Sacha Lodge, 250 m, 26–28.x.2004, FIT, G. de Rougemont leg. (1 ♀ CRO); "Peru: Dept. Cusco: Cock of the Rock Lodge, NE Paucanambo, 13°03.5′S 71° 32.7′W [−13.05833, −71.545], 1120 m, 4–9.xi.2007, D. Brzoska, ex. flight intercept trap, PER1B07 001" / "SEMC0871107" (1 ♂ SEMC); "Peru: Cuzco Dept., Consuelo, Manu Rd km 165, 9.x.1982" / "FMHD #82-361, beating dead branches, L.E. Watrous & G. Mazurek" (1 ♂ FMNH); same locality labels except 4.x.1982, FMHD #82-337, leaf litter (1 ♀ FMNH); same locality labels except 5.x.1982, FMHD #82-410, rotten palm bait trap (1 ♂ FMNH); same locality labels except 6–7.x.1982, FMHD #82-411, flight intercept trap (1 ♂ FMNH); same locality labels except Pillahuata, Manu Rd. km 128, 20.ix.1982, FMHD #82-265, litter along gravel stream (1 ♂ FMNH); same locality labels except Pillahuata, Manu Rd. km 128, 24.ix.1982, FMHD #82-283, litter along stream (1 ♂ FMNH); same locality labels except Pillahuata, Manu Rd. km 128, 27.ix.1982, FMHD #82-310, litter in runoff in mossy forest (1 ♂, 1 ♀ FMNH); same locality labels except Pillahuata, Manu Rd. km 128, 28.ix.1982, FMHD #82-311, litter along gravel stream (2 ♂ FMNH);"Peru: CU[sco] Camparmento Comerciato, 23.xi.2002, 12°47′S 73°22′W [−12.78333, −73.36666], 1,350 m, Pitfall, J. Grados" / "*Isanopus* spp. det. Asenjo 2004" (1 ♂ MUSM); "Peru: CU[sco] Camparmento Segakiato, 10.xi.2002, 12°43′S 73°18′W [−12.716666, −73.3], 1,850 m, Pitfall, J. Grados" (1 ♀ MUSM); "Peru: Dept. Loreto, Camparmento San Jacinto, 2°18.75′S 75°51.77W [−2.3125, −75.862833], 7.vii.1993, 175–215 m, R. Leschen #44, ex: flight intercept trap" / "SM0080093" (1 ♀ SEMC); "Peru: Dept. Loreto, 1.5 km N. Teniente Lopez, 2°35.66′S 76°06.92′W [−2.594333, −76.115333], 18.vii.1993, 210–240 m, R.

Leschen, #119. ex: flight intercept trap" / "SM0080094" (1 ♀ SEMC); "Peru: JU[nín], 1 km S Mina Pichita, 2100 m, 25.i.2005, 11°05′40.2″S 75° 4′49.6″W [−11.0945, −75.080444], A. Asenjo" (1 ♀ MUSM); "Peru, Dept. Madre de Dios: Pantiacolla Lodge, Alto Madre de Dios R., 12°39.3′S 71°13.9W [−12.655, −71.231666], 420 m, 14–19.xi.2007, D. Brzoska, ex. flight intercept trap, PER1B07 004" / "SEMC0872413" (1 ♂ SEMC); "Peru: Madre de Dios: Pantiacolla Lodge, 8 km NW El Mirador Trail, Alto Madre de Dios River, 800 m, 12°38′30″S 71° 16′41″W [−12.64166, −71.278055], 23–26.x.2000, R. Brooks, PERU1B00 102, ex: flight intercept trap" / SM0210891, SM0210653 (1 ♀ SEMC; 1 ♀UTCI); "Peru, Madre de Dios Dept., CICRA Field Station, trail 6, research plot, 12.55207°S 70.10962°W, 295 m, 11–13.vi.2011, Chaboo team, Malaise trap, PER-11-MAT-021" / "SEMC1060728" (1 ♂ SEMC); "Peru, Madre de Dios Dept., CICRA Field Station, ~2 km NW of cafeteria, research plot, 12.55212°S 70.10921°W, 295 m, 7–9.vi.2011, Chaboo team, flight intercept trap trap, PER-11-FIT-021" / "SEMC0956719" (1 ♀ SEMC); "Peru: Dept. Madre de Dios, Manu Prov., Parque Nac. Manu, Zona Res. Rio Manu, Cocha Juarez, trail nr. Manu" / "Lodge, 18–24.ix.1991, flight intercept trap, A. Hartman, Field Museum" (1 ♂ FMNH); "Peru: Madre de Dios, Tambopata Wildlife Res. 30 km SW Pto. Maldanado, 12°50′S 69°20′W [−12.83333, −69.3333], 290 m, 26.xi.1982, J.J. Anderson coll." (1 ♂ CMNH); "Peru, Ucauali Dept., Tingo Maria-Pucallpa Rd., Ruente Chino, km 205, 1,300 m, 9°8′12″S 75°47′20″W [−9.13666, −75.788888], 11–14.x.1999, R. Brooks, PERU1B99 007A, ex: flight intercept trap" / SM0185071, SM0185076 (1 ♂ SEMC; 1 ♂ UTCI). All paratypes with label "PARATYPE *Lendatus platys* Chatzimanolis, des. Chatzimanolis 2019".

**Diagnosis.** *Lendatus platys* and *L. bolivianus* can be distinguished from *L. philothalpiformis* by the coloration of head and pronotum (dark brown to black in *L. bolivianus* and *L. platys*; bright reddish-orange in *L. philothalpiformis*). *Lendatus platys* can be distinguished from *L. bolivianus* by the shape of the pronotum (becoming wide (convex) posteriorly (Fig. 2C) in *L. platys*; becoming narrower (concave) posteriorly (Fig. 2A) in *L. bolivianus*;); the shape of the paramere (paramere narrower, parallel-sided from base to apex in dorsal view (Fig. 7B) in *L. platys*; paramere wider, converging to apex in dorsal view (Fig. 5B) in *L. bolivianus*;) and the length comparison between the anterior portion of the paramere and median lobe (median lobe much longer than paramere (Figs. 7A–7B) in *L. platys*; median lobe slightly longer than paramere (Figs. 5A–5B) in *L. bolivianus*).

**Description.** Forebody length 4.7–5.6 mm. Coloration of head, pronotum and ventral side of body dark brown to black; mouthparts and antennae dark orange to brown; elytra metallic blue, green or purple (blue most commonly); legs dark brown except tarsi dark orange; abdomen dark brown to black except segment 7 (posterior 1/3 orange) and segment 8 (orange).

Head with 2–3 irregular rows of medium-sized punctures on each side of central impunctate area (except anteriorly); with additional 4–6 large punctures on epicranium; with microsculpture and micropunctures. Head width/length ratio = 1.53. Pronotum width/length ratio = 1.02; pronotum widest medially, lateral sides of pronotum convex; diagonal longitudinal line of punctures on disc of pronotum with 5–6 large punctures; anterolateral to that line pronotum with 5–8 medium-sized punctures; posterolateral to

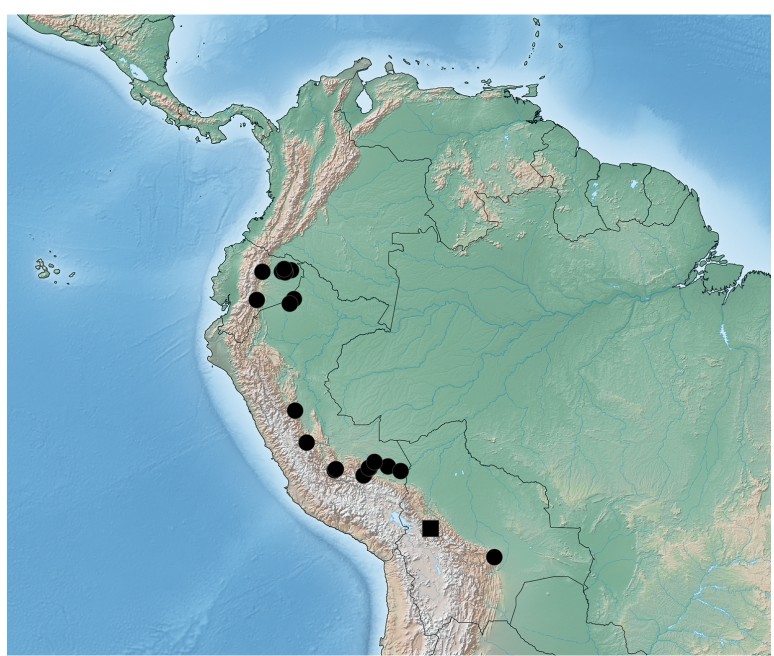

**Figure 8** Distribution map of *Lendatus bolivianus* Chatzimanolis (square) and *Lendatus platys* Chatzimanolis (circles).

that line pronotum impunctate; pronotum with microsculpture and micropunctures; pronotum/elytra length ration = 0.89. Males with narrow, deep emargination on sternum 7; sternum 8 with deep U-shaped emargination.

Aedeagus as in Fig. 7; paramere in dorsal view almost parallel-sided but apex slightly wider; in lateral view paramere convex, converging to broadly rounded apex; paramere with peg setae as in Fig. 7C; paramere narrower but longer than median lobe; median lobe in dorsal view converging to apex; in lateral view median lobe becoming narrower from middle to narrowly elongate apex; with small dorsal subapical tooth.

**Distribution.** Known from the department of Santa Cruz in Bolivia, the provinces of Morona-Santiago, Napo, Orellano and Sucumbios in Ecuador, and the departments of Cusco, Loreto, Junín, Madre de Dions and Ucauali in Peru.

**Habitat.** Specimens were collected with malaise, baited pitfall and flight intercept traps and by shifting leaf litter in wet tropical lowland forests or tropical cloud forests at elevations of 10–1,300 m).

**Etymology.** The specific epithet is derived from the Greek word $\pi\lambda\alpha\tau\upsilon\varsigma$ (wide) and refers to the wide shape of the pronotum.

**Remarks.** An additional specimen from Colombia [Colombia: Valle del Cauca, PNN Farallones de Cali, Anchicaya, 3°26′N 76°48′W, 730 m, 27.ii.–27.iii.2001, Malaise, S. Sarria leg., M1538" / ''SM0548730''(1 ♀ SEMC)] looks almost identical to this species, except that the pronotum is not as wide as the other specimens in this species. Unfortunately, this specimen is female and thus I cannot place it with certainty in *L. platys*.

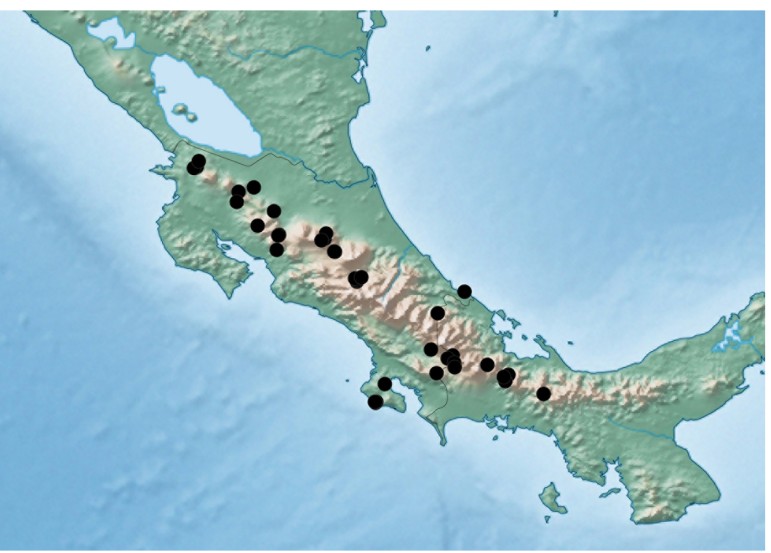

**Figure 9** Distribution map of *Lendatus philothalpiformis* Chatzimanolis.

## Key to the species of *Lendatus*

1.    Color of head and pronotum (Fig. 2B) bright reddish-orange (rarely brown); distributed in Central America (Fig. 9) ........................... *Lendatus philothalpiformis*

-    Color of head and pronotum dark brown to black (Figs. 2A, 2C); distributed in South America (Fig. 8) ........................................................................ 2

2.    Pronotum becoming narrower (concave) posteriorly (Fig. 2A); paramere wider, converging to apex in dorsal view (Fig. 5B); anterior portion of median lobe slightly longer than paramere (Figs. 5A–5B) ........................... *Lendatus bolivianus*

-    Pronotum becoming wide (convex) posteriorly (Fig. 2C); paramere narrower, parallel-sided from base to apex in dorsal view (Fig. 7B); anterior portion of median lobe much longer than paramere (Figs. 7A–7B) .............. *Lendatus platys*

## ACKNOWLEDGEMENTS

I thank all the curators and collection managers listed in the materials examined section for access to the specimens. I thank Adam Brunke, Aslak Kappel Hansen and an anonymous reviewer for comments that improved this manuscript. I thank Max Marlowe for taking some of the photographs shown in this paper.

### Funding

The author received no funding for this work.

## Competing Interests

The author declares there are no competing interests.

## Author Contributions

- Stylianos Chatzimanolis conceived and designed the experiments, performed the experiments, analyzed the data, contributed reagents/materials/analysis tools, prepared figures and/or tables, authored or reviewed drafts of the paper, approved the final draft.

## Data Availability

The 181 specimen accession and locations are available in the Supplemental File.

## New Species Registration

The following information was supplied regarding the registration of a newly described species:

Publication LSID: urn:lsid:zoobank.org:pub:0612FF19-38E8-4072-AF74-0EB16165841F

Lendatus LSID: urn:lsid:zoobank.org:act:73EEC4F3-E35B-4E67-9FAA-5C8C14222ABB

Lendatus bolivianus LSID: urn:lsid:zoobank.org:act:14E6C64D-E882-41D3-85DA-75930F62DCF1

Lendatus philothalpiformis LSID: urn:lsid:zoobank.org:act:7AFD3EE5-49B1-495D-A289-2C390B06BF61

Lendatus platys LSID: urn:lsid:zoobank.org:act:920D79A5-D1D4-4B73-8263-30D4908E3823.

## Supplemental Information

Supplemental information for this article can be found online at http://dx.doi.org/10.7717/peerj.7947#supplemental-information.

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
