# Peer review of "Lendatus, a new genus of Xanthopygina (Coleoptera: Staphylinidae: Staphylininae) with description of three new species"

_PeerJ, doi:10.7717/peerj.7947_

## Round 0.1 · original submission · Minor Revisions

Three reviewers have taken a careful look at your manuscript all of which agree that it is well written and nicely illustrated. However, all three do make some very nice suggestions that could be used to improve the paper (albeit an already very good manuscript). Generally speaking I agree with most of the comments but am unsure that another key is needed (as suggested by Reviewer 2). Some additional labels on your figures could improve clarity.

I would like to ask that you revise the paper with careful attention to each of the reviewer's comments. In your cover letter please document how each criticism/suggestion was addressed; in instances where you disagree with the reviewer (or feel the change is unwarranted), please provide justification for disregarding.

Finally, I think it's worth noting that lat/long data tend to be more portable in decimal degree format rather than degrees-minutes-seconds. I realize that could entail more work than necessary but on the other hand you might consider uploading the data to GBIF (if GBIF takes DMS and converts to DD - problem solved).

·

Basic reporting

The manuscript succeeds in all these areas of basic reporting.

Experimental design

The manuscript succeeds in all these areas of experimental design.

Validity of the findings

The manuscript succeeds in all areas of these standards.

Additional comments

Overall the manuscript is clear, concise and well-illustrated. I have some minor linguistic corrections in the attached file and some important (but simple) suggestions for improvement:

-in the abstract, it is said that the genus can be identified using a character on the male genitalia. I suggest swapping this for the very distinctive external character found in both sexes (pronotal punctation); I would also emphasize this elsewhere in the paper to clarify the message that it is actually easy to recognize the genus
-for L. platys: there is a single record on the west slopes on the Andes and this is a single female. I think this should be commented (whether there is any doubt about this record)

·

Basic reporting

English is good, sufficient references for the topic, good structure and relevant figures, tables and data accessibility.

My only structural comment would be the diagnosis sections, which to me fell more like comparison sections. I like having a diagnosis section containing the unique combination of characters within the genus and a separate comparison section focusing on the differences between the species and potential similar species (both within genus and in other genera). The diagnosis can also help define new taxa, as this combination can be compared to a potential new taxa.

Experimental design

For a taxonomic revision the article follows current traditions and is up to data in methodology.

Validity of the findings

Taxonomic revision are of high importance, especially in Neotropical settings, where much is still unrevised and unknown. This will bring much needed and useful information on new taxa from a poorly known region. It will be of importance for many other fields, amongst others, ecological assessments and biodiversity studies.

Additional comments

Only very minor comments:

* Would add a key to Isanopus group of genera
* Abdominal segments are usually termed in Roman numerals, e.g. segment V not 5.
* I have added a few comments, missing italics, etc. to attached pdf.

Reviewer 3 ·

Basic reporting

1. The author of the article has English as a native language.
2. The literature in the manuscript is referenced appropriately, the "Introduction" section is sufficient enough.
3. The whole article has a proper acceptable structure format. All the pictures are relevant to the content.
4. The obtained results of the article are relevant and confirm the previous hypotheses of the author.

Experimental design

The questions aroused in the article are meaningful and relevant for the modern systematics of the discussed group of beetles. The manuscript is well illustrated by original images. No ethical issues within the article. The methods used by Author are well described in the "Materials & Methods" section.

Validity of the findings

The manuscript is dedicated to description of a new genus and three species of the mega diverse beetle family Staphylinidae. The Author bring clear evidences for delimitation of the new genus and species. The article meets the PeerJ criteria and should be accepted as is.

Additional comments

Dear Stylianos,

It was a pleasure for me to make a review of your manuscript. I have only a minor comment. Could you please add markings in the pictures which indicate specific parts mentioned in the manuscript? It may help some readers to navigate the discussed structures.

All the best!

Annotated reviews are not available for download in order to protect the identity of reviewers who chose to remain anonymous.

---

## Round 0.2 · accepted · Accept

Thanks for you careful attention to my and the reviewers' comments. The manuscript looks great!